# Image Text Deblurring Method Based on Generative Adversarial Network

**Chunxue Wu** [1] , **Haiyan Du** [1], **Qunhui Wu** [2] **and Sheng Zhang** [1],*

[1]  School of Optical-Electrical and Computer Engineering, University of Shanghai for Science and Technology, Shanghai 200093, China; wcx@usst.edu.cn (C.W.); dhy5328@126.com (H.D.)

[2]  Shanghai HEST Co. Ltd. Shanghai 201610, China; shhest@aliyun.com

*  Correspondence: zhangsheng_usst@aliyun.com; Tel.: +86-13386002013

**Abstract:** In the automatic sorting process of express delivery, a three-segment code is used to represent a specific area assigned by a specific delivery person. In the process of obtaining the courier order information, the camera is affected by factors such as light, noise, and subject shake, which will cause the information on the courier order to be blurred, and some information will be lost. Therefore, this paper proposes an image text deblurring method based on a generative adversarial network. The model of the algorithm consists of two generative adversarial networks, combined with Wasserstein distance, using a combination of adversarial loss and perceptual loss on unpaired datasets to train the network model to restore the captured blurred images into clear and natural image. Compared with the traditional method, the advantage of this method is that the loss function between the input and output images can be calculated indirectly through the positive and negative generative adversarial networks. The Wasserstein distance can achieve a more stable training process and a more realistic generation effect. The constraints of adversarial loss and perceptual loss make the model capable of training on unpaired datasets. The experimental results on the GOPRO test dataset and the self-built unpaired dataset showed that the two indicators, peak signal-to-noise ratio (PSNR) and structural similarity index (SSIM), increased by 13.3% and 3%, respectively. The human perception test results demonstrated that the algorithm proposed in this paper was better than the traditional blur algorithm as the deblurring effect was better.

**Keywords:** image deblurring; generative adversarial network; Wasserstein distance; adversarial loss; perceptual loss

## 1. Introduction

Image restoration [1] is an important research direction in image processing. It is a technique to study the cause of degradation and establish a mathematical model to restore high-quality images in response to the degradation in the image acquisition process. Image deblurring [2,3] is also a kind of image restoration. It is mainly aimed at the blurring effect caused by the relative displacement between the photographed object and the device due to camera shake or noise interference. The texture is also clearer and more natural. With this technology, low-quality blurred images can be restored to high-quality clear images. At present, there are many methods applied in the field of image deblurring. However, due to the high-quality requirements of image deblurring, image deblurring is still a very challenging research direction.

With the continuous development of human industrial technology, the application of characters on workpieces in the industrial field is also very important. The on-site environment in an industrial site is chaotic and complex. Factors such as camera shake, subject movement, light, and on-site noise will cause the image captured by the camera to appear blurred. The integrity of the information is

urgently required in industrial sites. Solving this problem also has practical significance and application value. In the intelligent express sorting system, as long as the express code is obtained, you can know the area where the express was sent by the courier. If these courier slips are missing, guessing the information with the naked eye is not only cumbersome and heavy but also inefficient. If you can use a computer to automatically recover the blurred image information and automatically restore the blurred three-segment coding from the express order to a clear image, then you can reduce the manpower and material resources for manual data processing. Therefore, it is very important to deblur the fuzzy express image information in the automatic sorting system.

In recent years, with the rapid development of deep learning, a large number of scholars have researched image deblurring methods based on machine learning, and all have achieved good results [4]. In many studies, the image deblurring method based on generative adversarial network (GAN) has been widely recognized. This not only takes into account the rationality of image texture details but also considers the uniformity of the overall image structure. Therefore, this paper proposes an image deblurring method based on the generative adversarial network. Firstly, in order to eliminate the difference between the real blurred image and the generated blurred image, we established an unpaired dataset and solved the image deblurring problem based on this. Then, a GAN model was established, which consisted of two generative adversarial networks. For the conversion of blurred images to clear images and the conversion of clear images to blurred images, the loss function was optimized by combining adversarial loss and perceptual loss. Finally, a stable network model was obtained by iteratively training the GAN model on unpaired datasets. During each training process, the model was updated to achieve a better deblurring effect.

The specific sections of the article are as follows. Section 2 briefly introduces the related work on image deblurring. Section 3 presents the network model proposed in this paper and explains each part of the model in detail. Section 4 gives the experimental demonstration, and Section 5 gives the summary of this article and future expectations.

## 2. Related Works

With the popularization of handheld devices and multimedia communication technology, image deblurring technology avoids further increasing the high cost of the device as well as noise interference between redundant components. It has broad application scenarios and strong technical advantages. If we can first use the deblurring method to restore the corresponding clear image and then use the restored image as the input of the subsequent neural network, the accuracy of the output of the algorithm will be greatly improved. Therefore, image deblurring technology, as an important part of computer vision data preprocessing, has also become a research hotspot in the field of computer vision and computer graphics.

At present, there are two main research methods for image deblurring. One is the nonblind deblurring method, which uses a known blur kernel function that directly deconvolves the degraded model of the blurred image to obtain a restored high-definition image. The other is the blind deblurring method, which is used when the fuzzy process is unknown. A brief introduction of these two methods is as follows.

The nonblind deblurring method is a more traditional image deblurring method. It first obtains the blur kernel information through a certain technique and then deconvolves the blur image according to the obtained blur kernel to restore a high-definition image. The classic deconvolution algorithms include the Lucy–Richardson algorithm, the Wiener filter, and the Tikhonov filter.

In reality, in most cases, the fuzzy function is unknown. Therefore, it is necessary to make assumptions on the fuzzy source and parameterize the fuzzy function. The most common assumption is that the blur is uniformly distributed on the image. For example, the method proposed by Fergus et al. [5] achieved groundbreaking results, and the literature [6–8] has been optimized based on it. In addition, there are some methods for dealing with cases where the blur is unevenly distributed on the image, but this type of algorithm also simplifies the problem from different angles. For example,

Whyte et al. [9] used a parametric geometric model to model camera motion, and Gupta et al. [10] assumed that blur was caused solely by 3D camera motion. These traditional methods have achieved certain effects. However, because the model makes too many assumptions, they have a lot of limitations in the application scene and cannot solve the problem of image blur caused by various complicated factors in actual life.

With the development of deep learning in the field of computer vision, scholars everywhere have begun to use deep learning to deal with image deblurring. Earlier works were still based on the idea of nonblind deblurring, allowing neural networks to estimate fuzzy kernel information. For example, Sun et al. [11] used a convolutional neural network (CNN) to estimate the fuzzy kernel and then restored the image based on the estimated fuzzy kernel. Chakrabarti et al. [12] used de-CNN to predict the Fourier coefficients of the fuzzy kernel and deblurred the image in the frequency domain. Gong et al. [13] used a full convolutional network (FCN) to estimate the motion flow of the entire image and restored a blurred image based on it. Due to the use of a nonblind deblurring algorithm, the above methods need to obtain a clear image after obtaining the estimated fuzzy kernel through CNN and then use a traditional deconvolution algorithm to deconvolve the blurred image. This leads to slow running speed of the algorithm, and the restoration effect depends entirely on the estimation results of the blur kernel.

In recent years, with the deep prospect of deep learning in the areas of image semantic repair and image compression [14], more and more scholars have discovered that the work that neural networks can cover is far more than just estimating fuzzy kernels. In 2017, Nah et al. [15] proposed the use of multiscale convolutional neural networks to directly deblur images. They used an end-to-end training method to allow the network to directly reproduce clear images without first estimating the blur function. This type of method is called the blind deblurring method. Compared with the previous method, this method greatly improves the model effect and running speed. Other similar methods are those proposed by Noroozi et al. [16], Ramakrishnan et al. [17], and Yao et al. [18,19]. Later, Kupyn et al. [20] proposed the use of conditional generative adversarial networks (CGAN) to deblur images. They followed the basic structure of pix2pix, a general framework for image translation tasks proposed by Isola et al. [21], and modified it to obtain the DeblurGAN image deblurring algorithm model. This model obtained better image deblurring effect than the multiscale convolutional neural network used by Nah et al. At the same time, the network structure was simpler and faster. To some extent, this reflects the fact that the generative adversarial network really performs well on image deblurring tasks.

In this paper, an image deblurring method based on GAN is proposed for unpaired datasets. Because there is no blur–clear image pair in unpaired datasets, a single GAN cannot directly calculate the loss function. Therefore, the proposed model uses two generations. Adversarial networks can realize the mutual conversion between blur and clear images and indirectly calculate the loss function between the input and output images. Therefore, the model has the ability to train and learn on unpaired datasets. At the same time, a loss function that combines adversarial loss and perceptual loss is used for training, making the image generated by the model clearer and more real.

## 3. Image Deblurring Model

In order to eliminate the difference between the real blurred image and the blurred image synthesized by the algorithm, as well as to achieve a better image deblurring effect in the real industrial scene, we used the CycleGAN structure [22] on the premise of having an unpaired dataset. An image deblurring model based on a generative adversarial network was established. The overall structure of the model is shown in Figure 1. This model consists of two generative adversarial networks A and B, which are used to achieve the conversion from blurred images to clear images and from clear images to blurred images. A and B networks are composed of their respective generators and discriminators. The model also adds a loss function to the network that combines adversarial loss and perceptual loss. Such a model just constitutes a cyclic structure consisting of clear-> blur-> clear and blur-> clear->

blur, which better constrains the content of the generated sample. This article will introduce each part of the model separately.

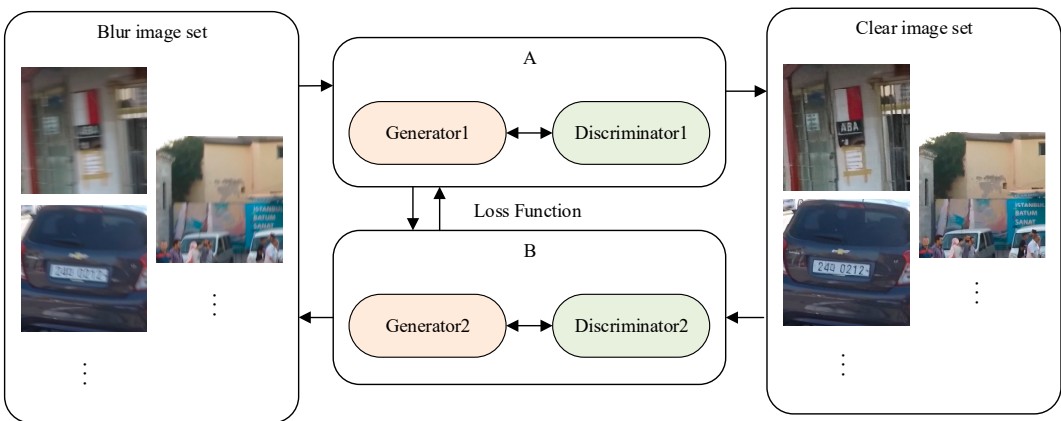

**Figure 1.** The overall structure of the model.

*3.1. Structure of GAN*

### 3.1.1. Generator Model

In generative adversarial networks, generative networks are the key. In previous studies, some classic network structures [23,24] have also achieved outstanding results. Among them, the deep residual network structure has achieved better performance in generating high-definition image tasks because it can greatly increase the number of layers in the network, as shown in [25,26]. The deep residual network was proposed by He et al. [24] in 2016. It is composed of several residual block (ResBlock) and other layers. After using the residual module, the number of layers of the network can be greatly deepened without the phenomenon of difficult model convergence, model degradation, and disappearance of gradients. Therefore, we used the deep residual network structure as the network structure of the generator. In addition, we added global skip connection [27] to the generative network. Global skip connection can directly connect the input of the entire network to the output. Therefore, the intermediate network only learns the residual between the output and the input, thereby reducing the amount of network learning and making the network converge faster and fit to improve the generalization ability of the model.

On the specific network structure, combined with CycleGAN, deep residual network, and global skip connection, the network contains 22 convolutional layers and two deconvolutional layers. A batch normalization (BN) layer [28,29] is added after each convolutional layer. The activation function uses a linear rectification (ReLU) function. The network structure of the generator is shown in Figure 2.

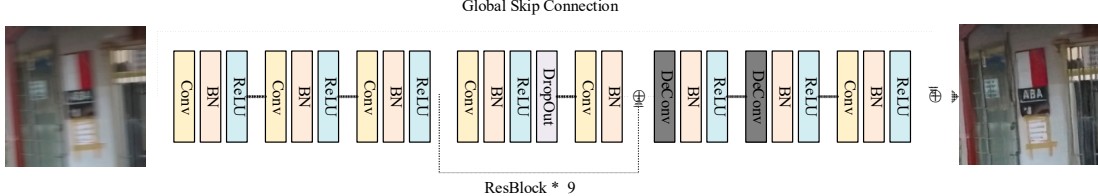

**Figure 2.** Generator network.

### 3.1.2. Discriminator Model

Compared with the generator, the task of the discriminator is to learn the difference between the generated sample and the real sample, including whether the structure in the picture is natural and the content is clear. In order to more accurately measure whether the samples are clear and natural,

we used PatchGAN, proposed by Isola et al. [21], as the discriminative network structure. Because PatchGAN pays more attention to the local information of the image, the generated image details are more abundant, and the visual effect is more realistic. Unlike the original PatchGAN, we removed the last Sigmoid function activation layer of the original PatchGAN and used Wasserstein distance instead of the original loss function. PatchGAN is unique in that it pays more attention to the local information of the image, which makes the generated image richer with detailed information and the visual effect more realistic.

Because the input of PatchGAN is an image block, we used a sliding window with a size of $70 \times 70$ to traverse the entire generated sample. Each image block can output a value through PatchGAN, and the average value of all image blocks can then be obtained with the authenticity of the entire image. The structure of the entire discrimination network is shown in Figure 3.

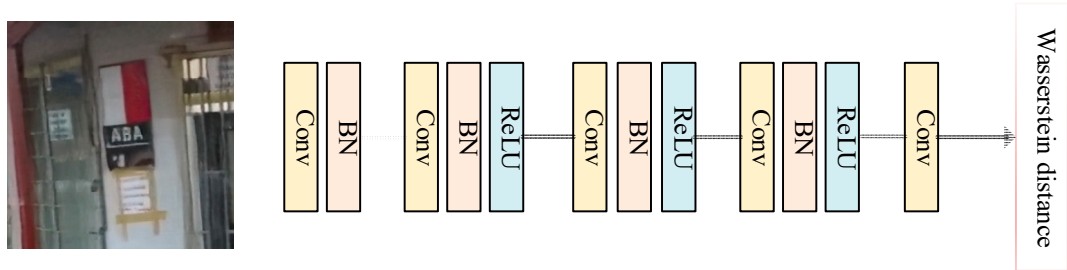

**Figure 3.** Discriminator network.

### 3.2. Loss Function of Network

The loss function is the most basic and critical factor in deep learning. By rationally designing the loss function and continuously optimizing it, the network can learn what they should learn without a clear image, thereby achieving a deblurring effect. In this work, the loss function of the entire network is a combination of adversarial loss [30] and perceptual loss. With these two loss functions, the generator can produce clear and realistic images. For the convenience of description, in the following content, Z is used to represent the samples in the clear image set, T is the samples in the blur image set, and N is the number of samples.

#### 3.2.1. Adversarial Loss

Adversarial loss refers to the loss function between two generative adversarial networks A and B. For A, its role is to make the generated image as realistic and clear as possible, and for B, its role is to make the generated sample have as realistic motion blur as possible. In the development of generative adversarial networks, various adversarial loss functions have appeared, including cross-entropy loss functions [31], squared loss functions [32], and Wasserstein distance loss functions [33]. Because WGAN-GP [33,34] uses the Wasserstein distance loss function as the adversarial loss of the network and increases the gradient penalty term for discriminating the network, it has achieved the most stable training effect at present. Therefore, we used the Wasserstein distance loss function for the confrontation loss. The calculation process of the adversarial loss is shown in Figure 4.

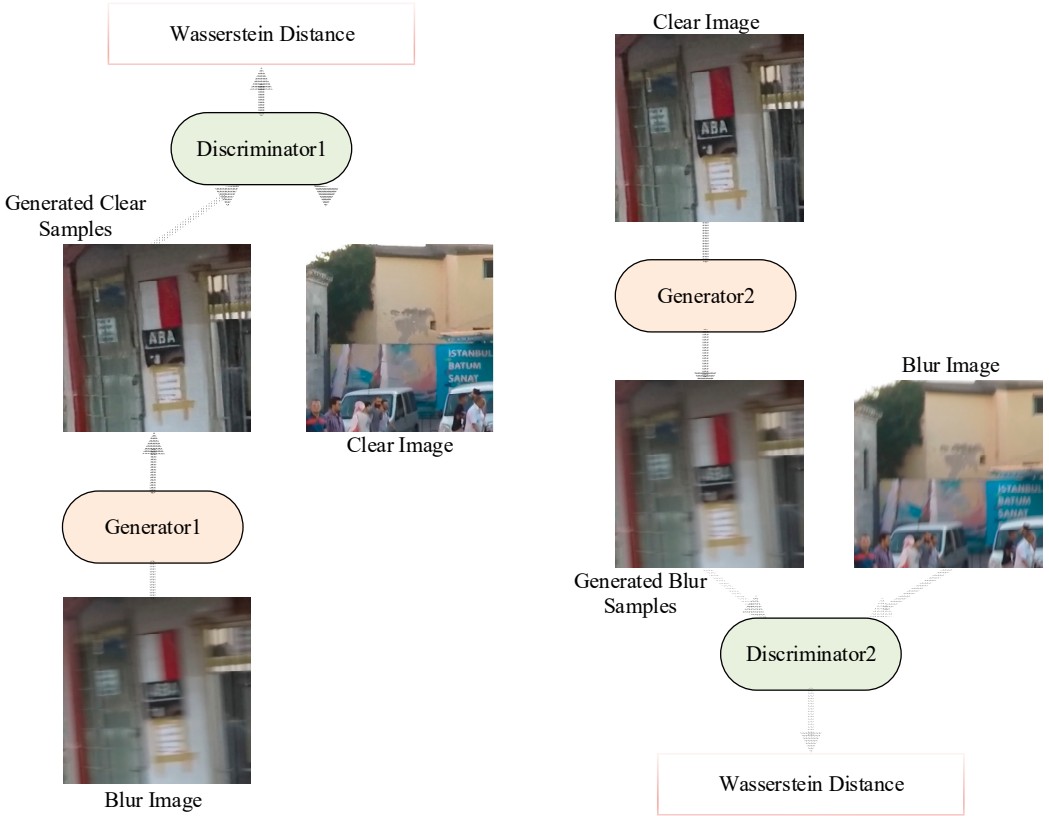

**Figure 4.** The calculation process of adversarial loss.

The formulas of the two generative adversarial networks are shown in Equations (1) and (2).

$$L_{GAN}(A) = \frac{1}{N} \sum_{n=1}^{N} [D_A(T) - D_A(G_A(Z))] \tag{1}$$

$$L_{GAN}(B) = \frac{1}{N} \sum_{n=1}^{N} [D_B(Z) - D_B(G_B(T))] \tag{2}$$

In the above formula, $G_A(Z)$ represents the sample generated by the generator in network A on the clear image set Z, and $G_B(T)$ represents the sample generated by the generator in network B on the blurred image set T. $D_A(T)$ represents the probability that the discriminator in network A judges whether the blurred image set T is a real image. $D_B(Z)$ represents the probability that the discriminator in network B judges whether the clear image set Z is a real image.

3.2.2. Perceptual Loss

Perceptual loss has the ability of visual perception close to the human eye. Compared with other pixel-level loss functions, it can make the generated image look more realistic and natural. Perceptual loss was originally proposed by Johnson et al. [35], and it has achieved good results in multiple application areas, such as image style transfer [35], image segmentation [36,37], image super-resolution reconstruction [26,38], and image deblurring [20]. The calculation of the perceptual loss depends on the visual geometric group (VGG) network [39]. The specific calculation steps are as follows. First, input two images to be tested, namely, a real image and a generated image, into a pretrained VGG network. Then, extract the feature map output by one or several convolutional layers from the VGG network. Finally, calculate the mean square error (MSE) on the feature maps corresponding to the two images to be tested. In this work, the feature map output from the eighth convolution layer in the

VGG-16 network was selected to calculate the perceptual loss. The calculation process is shown in Figure 5. The formula is shown in Equation (3).

$$L_{per}(G_A, G_B) = \frac{1}{N} \sum_{n=1}^{N} \left[ \frac{1}{shw} \varphi(G_B(G_A(Z))) - \varphi(Z)_2^2 + \frac{1}{shw} \varphi(G_A(G_B(T))) - \varphi(T)_2^2 \right] \quad (3)$$

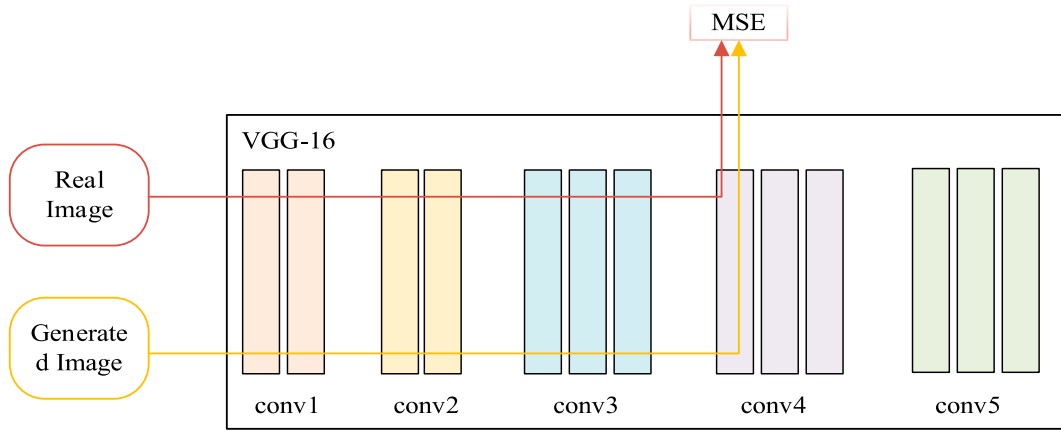

**Figure 5.** The calculation process of perceptual loss.

Here, $\varphi$ represents the feature map output from the eighth convolutional layer of VGG-16, and $s$, $h$, and $w$ represent the number of channels, height, and width of the feature map, respectively.

In summary, the overall loss function of the network is the result of the weighted summation of the above two loss functions, and the formula is shown in Equation (4).

$$L(A, B, Z, T) = L_{GAN}(G_A, D_A, Z, T) + L_{GAN}(G_B, D_B, T, Z) + \mu L_{per}(G_A, G_B) \quad (4)$$

In the above formula, $\mu$ represents the weight of the perceptual loss function. As with the original generative adversarial network, the generator needs to minimize the loss function, and the discriminator needs to maximize the loss function. Therefore, the result of the optimal generator is as follows:

$$G_A', G_B' = arg \min_{G_A, G_B} \max_{D_A, D_B} L(A, B, Z, T) \quad (5)$$

### 3.3. Algorithm Implementation

In the algorithm implementation process of the entire network model, the role of Discriminator1 and Discriminator2 is to provide gradients for Generator1 and Generator2 to guide its optimization process. The role of generative network B is to realize the conversion from clear images to blurred images and assist generative network A to complete the learning process so that A can generate and input consistent samples. After Discriminator1, Generator2, and Discriminator2 complete their respective auxiliary work, Generator1 is responsible for restoring the input blurred image into a clear image as the output result of the entire algorithm model. In the model training process, we used the buffer pool strategy proposed by Shrivastava et al. [40] to reduce model oscillation. When updating the parameters of the discriminant network, the historical samples generated by the generator and the new samples generated in this iteration were used as the input of the discriminative network, thereby increasing the stability of the model. The size of the generated sample buffer pool was 50, the batch size of all model training was 1, and the number of iterations was 300 epochs. The optimization algorithm used the Adam algorithm [41], and the initial learning rate was 0.0002. The specific implementation process was as Algorithm 1.

---

**Algorithm 1:** The algorithm flow of this model.

---

1: Initialize the input shape, h = 256, w = 256, s = 3, and the output shape of PatchGAN, pa = h/2**4, and the loss weight, $\mu$ = 10, and the optimizer, Adam(0.0002, 0.5)

2: Input (img_1,h,w,s), Input (img_2,h,w,s)

3:    Combined model trains generator to discriminator

4:    **for** epoch **in** range(300):

5:    **for** batch_i, img_1, img_2 **in** enumerate (dataloader.loadbtach(1)):

6:      fake_2 = generator(img_1), fake_1 = generator(img_2)

7:      recon_1 = generator(fake_2), recon_2 = generator(fake_1)

8:      vali_1 = discriminator(fake_1), vali_2 = discriminator(fake_2)

9:    **if** loss **is** $arg \min_{G_A,G_B} \max_{D_A,D_B} L(A, B, Z, T)$**:**

10:    clear_image = concatenate (img_1, fake_2, recon_1, img_2, fake_1, recon_2)

11: **return** clear_image

---

## 4. Experiments and Results

The GAN model proposed in this paper is implemented based on the Python language and the PyTorch deep learning framework, and it can run on an Intel Xeon computer with a 2.40 GHz CPU and 32 GB RAM. The model idea was established by looking at the references for one month. The model training took 15 days, and the model finally reached a stable state.

### 4.1. Datasets

Because there are no large-scale unpaired image deblurring datasets yet to be disclosed, it is meant for comparison with algorithms on paired datasets. Therefore, we still used paired datasets for training but used unpaired training methods. We used two datasets: the GOPRO dataset and a small unpaired dataset built in this work. All models were trained on the GOPRO dataset. The self-built unpaired dataset was only used as a test dataset due to its small number. These two datasets are described separately.

#### 4.1.1. GOPRO Dataset

The GOPRO dataset is currently the largest and the highest-resolution open-paired dataset in the field of image deblurring. It consists of 3214 pairs of blurred and clear images. In order to obtain a more realistic blurred image, all the blurred images in this dataset are fused from multiple clear images taken in the real scene rather than synthesized by means of clear image convolution blur kernel. For this work, a simplified version of this dataset was downloaded from the internet. The dataset was used as the training dataset, and the images in all datasets were cropped from the original GOPRO dataset. The size was 256 × 256. An image of a partial dataset is shown in Figure 6.

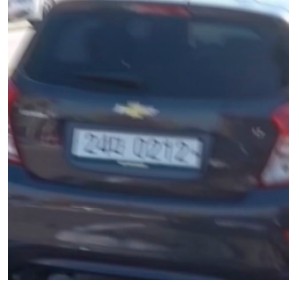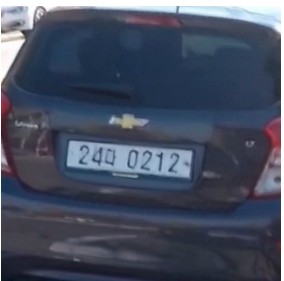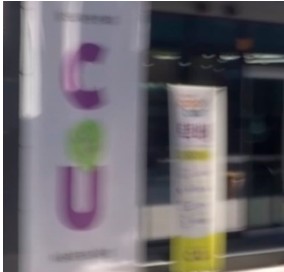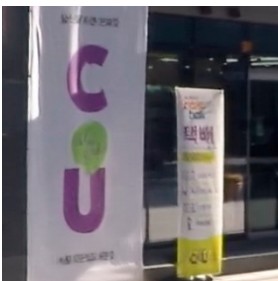

**Figure 6.** Partial GOPRO dataset.

4.1.2. Unpaired Dataset

In order to eliminate the difference between the real blurred image and the blurred image synthesized by the algorithm, as well as to achieve a better image deblurring effect in the real industrial scene, we trained the network model by unpaired training. In order to test the effectiveness of the model in real scenarios, a small unpaired test dataset was also established. The data contained only 70 blurred images, which were collected or photographed by the author from different sources, and they all originated from real scenes. An image of a partial dataset is shown in Figure 7.

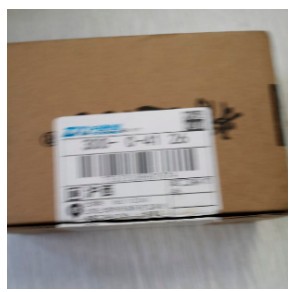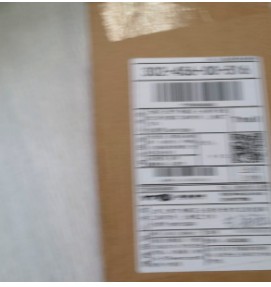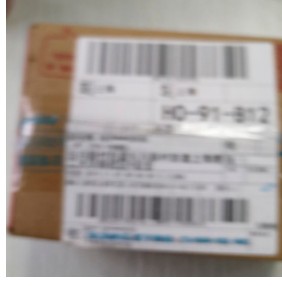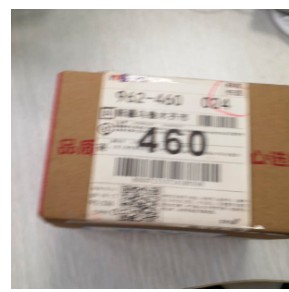

**Figure 7.** Partial unpaired dataset.

*4.2. Experimental Results and Comparison*

Based on the CycleGAN, we established an image deblurring algorithm model based on a generative adversarial network. First, the model was trained on the GOPRO dataset, and the number of iterations was 300 epochs. Then, in order to verify the validity of this model, the CycleGAN and DeblurGAN methods were selected to make comparisons with the algorithm used in this work, and they were tested on GOPRO test dataset and self-built unpaired dataset. In order to objectively and comprehensively analyze the deblurring effect of the algorithm, we used two quantitative evaluation indicators of peak signal-to-noise ratio (PSNR) and structural similarity index (SSIM) as well as human perception test methods for evaluation. PSNR evaluates the quality of an image by measuring the error between corresponding pixels in two images, while SSIM measures the degree of image distortion from brightness, contrast, and structural information. The human perception test refers to evaluation of the test results of the model with the human eye. The test subject needs to choose the one that is considered better from the two given images or indicate whether the choice is uncertain. The two images given were reconstructed from two randomly selected models among the three contrasting models. After the subjects completed the test, we combined the pass rate and the TrueSkill evaluation system to measure the deblurring ability of the three models.

First, this paper will show the test results of each model on the GOPRO test dataset. Table 1 shows the quantitative evaluation results, Figure 8 shows the perceptual test results of each model on the GOPRO test set, and Figure 9 shows the deblurring results of each model on the GOPRO test set.

**Table 1.** Quantitative evaluation results on GOPRO dataset.

| Models〈Parameters | PSNR | SSIM |
|---|---|---|
| CycleGAN | 22.6 | 0.902 |
| Ours | 25.6 | 0.929 |
| DeblurGAN | 26.8 | 0.943 |

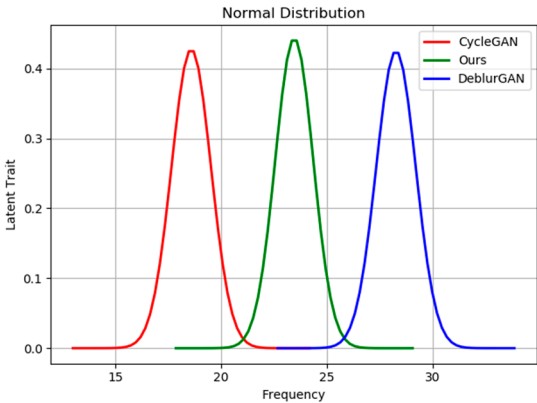

**Figure 8.** Perceptual test results on the GOPRO test set.

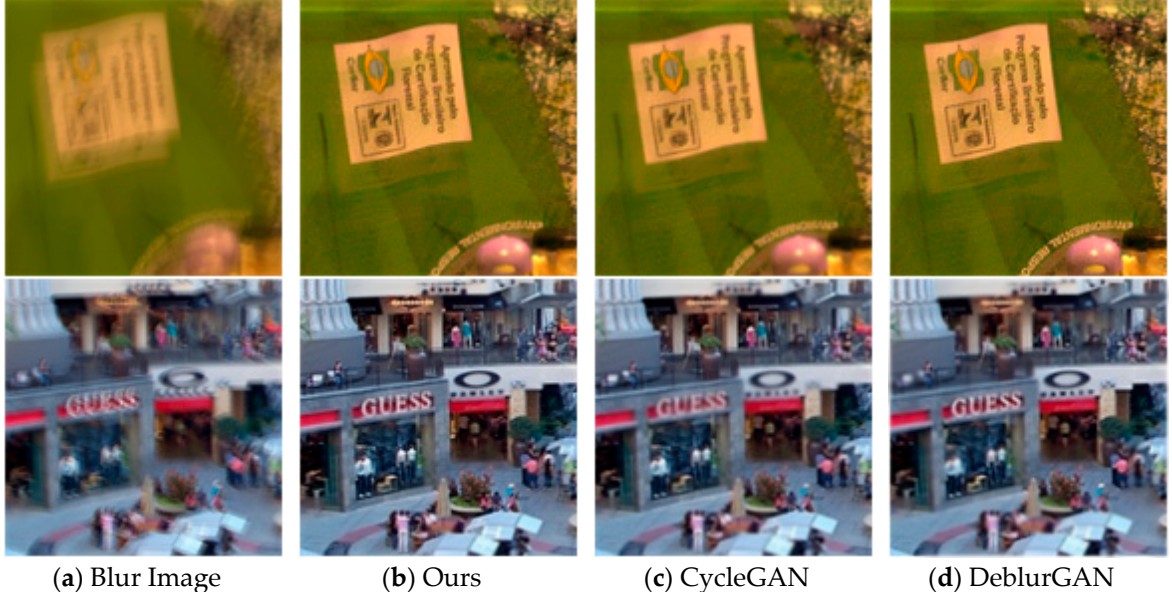

| (**a**) Blur Image | (**b**) Ours | (**c**) CycleGAN | (**d**) DeblurGAN |

**Figure 9.** Experimental results of different algorithms on GOPRO dataset.

As can be seen from Table 1, the results of the method used in this paper were significantly improved compared to CycleGAN, and the PSNR and SSIM indicators increased by 13.3% and 3%, respectively. Compared with DeblurGAN, our method achieved similar results.

In Figure 8, the results of each model are represented by a Gaussian curve. The mean $\mu$ of the curve represents the average deblurring ability of the model, and the variance $\sigma$ represents the stability of the model. The higher the average value of the curve and the smaller the variance, the better the model. Therefore, it can be seen from the above figure that the model in this paper had better deblurring effect compared to CycleGAN, but it was slightly worse than DeblurGAN.

From the results in Figure 9, it can be seen that, compared with CycleGAN, the effect of the proposed model was significantly improved. Not only was the deblurring ability enhanced, but the disadvantages of chromatic aberration, edge distortion, and unnatural sharpness were also eliminated, making the repair. The resulting image looked more real and natural. Compared with DeblurGAN, the method used in this paper obtained similar results on images with low blurring degree, and the images after deblurring were all natural and clear. However, on the images with a high degree of blurring, the deblurring effect of the method used in this paper was not thorough enough, and it was not as good as DeblurGAN.

Next, this paper will show the test results of each model on a self-built unpaired dataset. Figure 10 shows the perceptual test results of each model on the unpaired dataset, and Figure 11 shows the deblurring results of each model on the unpaired dataset.

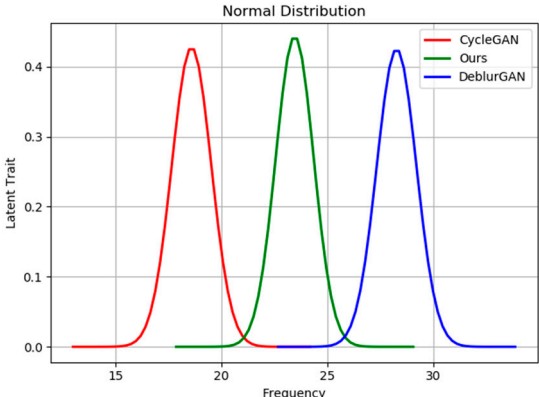

**Figure 10.** Perceptual test results on unpaired dataset.

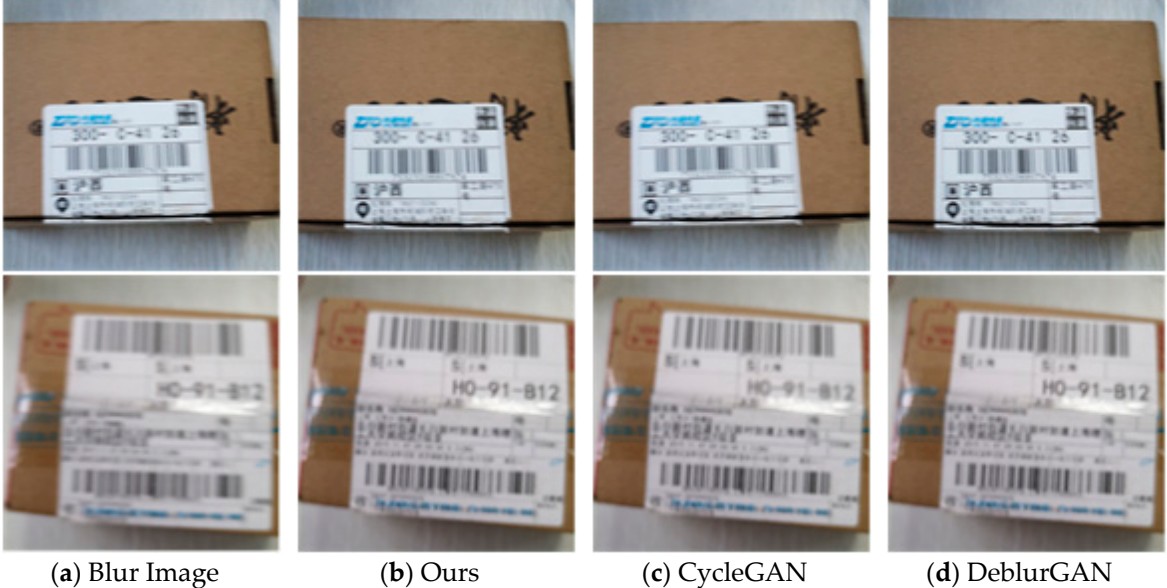

| (**a**) Blur Image | (**b**) Ours | (**c**) CycleGAN | (**d**) DeblurGAN |

**Figure 11.** Experimental results of different algorithms on unpaired dataset.

From the results in Figures 10 and 11, we can see that, on the unpaired dataset, the proposed model had better deblurring effect than CycleGAN, and the repaired image looked more realistic and natural. However, its performance was slightly worse than DeblurGAN.

Combining the experimental results on the above two datasets, it can be seen that the loss function, which combines the adversarial loss and perceptual loss, can play a certain role in constraining the content of the generated image. However, because the generated image was not directly constrained, the image was generated by the constraint. The reconstructed image was used to indirectly constrain the generated image, so the effect was limited, and the effect achieved was not as good as that on the paired dataset. However, in general, the method achieved certain results on highly difficult unpaired datasets. Compared to traditional CycleGAN, the deblurring effect was significantly improved. If a large-scale image dataset in a real scene can be obtained, the effect will be better.

In order to show the importance of the combination of various parts of the model to the deblurring effect, ablation research was also performed in this work. The ablation of the model was achieved by reducing the number of convolution layers of the generator network to 16 layers and removing one

of the perceptual loss functions. Finally, through model training, the evaluation index results on the GOPRO paired dataset are shown in Table 2.

**Table 2.** Quantitative evaluation results after ablation.

| Models | Parameters | PSNR | SSIM |
|---|---|---|---|
| Ours | | 25.6 | 0.929 |
| Ablation | | 18.5 | 0.767 |

As can be seen from the above table, after reducing the generator convolutional layer and removing the perceptual loss, the PSNR and SSIM evaluation indexes were reduced too much, by 27% and 17%, respectively, resulting in poor model performance. Therefore, the various components of the model proposed in this work are particularly important in the field of image deblurring.

## 5. Conclusions and Future Works

With the widespread use of handheld devices and digital images on multimedia communication networks, image deblurring technology has more and more application value. In particular, in an intelligent express sorting system, using a computer to restore the fuzzy three-segment coded information on the courier slip to a clear image can improve the recognition effect of the subsequent three-segment code. Therefore, this paper proposes an image deblurring method based on generative adversarial networks. First, in view of the shortcomings of the existing algorithms, we dealt with image deblurring on unpaired datasets to solve motion deblurring in actual scenes. Then, based on CycleGAN, an image deblurring model based on a generative adversarial network was established to realize the conversion of blurred images to clear images and the conversion of clear images to blurred images. The process of network learning was also constrained by combining adversarial loss and perceived loss so that the network could better learn the motion-blurred data distribution characteristics in the actual scene. Finally, an evaluation was performed on the GOPRO dataset and the self-built unpaired dataset. The experimental results showed that the proposed method could obtain good deblurring effect on both datasets and that it was better than CycleGAN. However, some improvements are still required. For example, in future work, we may try to introduce a multiscale network structure into the model and deepen the network layers at the same time to improve the capacity of the model. There are also loss functions of other structures designed to strengthen their constraints on the generated sample content, which can be used to achieve a complete deblurring effect.

**Author Contributions:** Conceptualization, C.W.; methodology, H.D. and Q.W.; software, H.D.; validation, H.D.; formal analysis, C.W. and H.D.; investigation, Q.W.; resources, S.Z.; data curation, H.D.; writing—original draft preparation, H.D.; writing—review and editing, H.D.; visualization, H.D.; supervision, S.Z. All authors have read and agree to the published version of the manuscript.

**Funding:** This research was funded by SHANGHAI SCIENCE AND TECHNOLOGY INNOVATION ACTION PLAN PROJECT, grant number 19511105103.

**Acknowledgments:** The authors would like to thank all anonymous reviewers for their insightful comments and constructive suggestions to polish this paper to high quality. This research was supported by the Shanghai Science and Technology Innovation Action Plan Project (19511105103) and the Shanghai Key Lab of Modern Optical System.

**Conflicts of Interest:** The authors declare no conflict of interest. The funders had no role in the design of the study; in the collection, analyses, or interpretation of data; in the writing of the manuscript, or in the decision to publish the results.

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
