# Peer review of "Image Text Deblurring Method Based on Generative Adversarial Network"

_electronics, doi:10.3390/electronics9020220_

Round 1

Reviewer 1 Report

1. Abstract. Please add some detail on experiments/datasets used and quantitative details on accuracies attained.
2. Abstract. please justify a bit more on novelty of the method, and relative benefits compared to
existing state-of-the-art approaches. Although justification is provided later, it will be beneficial to briefly state
why Wasserstein is used, say vs. other metrics.
3. line 110. do not list conference name, citation is sufficient
4. line 118. please expand and tell the reader what is the benefit of 2 adversarial network vs. one. Also,
briefly highlight the advantage of combining adversarial loss and perceptual loss.
5. line 120. algorithm or model? choose one
6. line 125. Why 2 adversarial networks, why not 3, 4,....? Please explain.
7. equations 3.1 and 3.2. Need to define all parameter variables in text so the reader is clear on what they represent.
8. can the authors please add some discussion on possible limitations of the method /algorithm
9. Please provide some indication in manuscript on training time and inference time.

Author Response

Point 1: Abstract. Please add some detail on experiments/datasets used and quantitative details on accuracies attained.

Response 1: The data set used in this paper and the results of the evaluation indicators reached on the data set have been added to the abstract. (in yellow)

Point 2: Abstract. please justify a bit more on novelty of the method, and relative benefits compared to existing state-of-the-art approaches. Although justification is provided later, it will be beneficial to briefly state why Wasserstein is used, say vs. other metrics.

Response 2: The benefits of the proposed method over traditional methods have been described in the abstract, including the advantages of using two generative adversarial networks, Wasserstein distance, and a loss function that combines adversarial and perceptual losses. (in yellow)

Point 3: line 110. do not list conference name, citation is sufficient.

Response 3: line 116. Conference name has been removed from this paper. (in yellow)

Point 4: line 118. please expand and tell the reader what is the benefit of 2 adversarial network vs. one. Also, briefly highlight the advantage of combining adversarial loss and perceptual loss.

Response 4: line 126. This paper has briefly explained the advantages of using 2 generative adversarial networks and the advantage of combining adversarial loss and perceptual loss. (in yellow)

Point 5: line 120. algorithm or model? choose one.

Response 5: line 131. This paper has been modified to an image deblurring model. (in yellow)

Point 6: line 125. Why 2 adversarial networks, why not 3, 4,....? Please explain.

Response 6: Because one GAN is used to convert the blurred image to a clear image, and the other GAN is used to achieve the conversion from a clear image to a blurred image. By connecting two independent blur and clear image domains, two loops can be formed, clear- > Blur-> Clear and Blur-> Clear-> Blur. Therefore, the calculation of the loss function under unpaired data can be realized. Since 2 GANs already form a cycle in both directions, there is no need to use 3 or more GANs.

Point 7: equations 3.1 and 3.2. Need to define all parameter variables in text so the reader is clear on what they represent.

Response 7: line 206. This paper has given a detailed explanation of the parameter variables used in equations 3.1 and 3.2. (in yellow)

Point 8: can the authors please add some discussion on possible limitations of the method /algorithm.

Response 8: line 338. This paper adds a discussion of the limitations of the method. (in yellow)

Point 9: Please provide some indication in manuscript on training time and inference time.

Response 9: line 256. This paper has added model training time and inference time. (in yellow)

Reviewer 2 Report

This paper proposes an image text deblurring method based on GAN. The motivation is clear, and the method is generally well presented. 

However, there are still some major concerns:

1. The authors claimed the method can deal with unpaired settings, but the experiments are based on paired ones. The authors can construct unpaired datasets based on existing ones. For example, paired (A1, B1), (A2, B2) can be reconstructed as (A1, B2), (A2, B1).

2. To my knowledge, training GAN is unstable. How to tackle this in this paper? Using cycle-consistent GAN might be a solution.

3. The ablations studies are missing on each component of the method.

4. Some closely related works are missing, such as "Text image deblurring via two-tone prior", "Towards more efficient and flexible face image deblurring using robust salient face landmark detection" on image deblurring.

5. Careful proofreading is required.

Author Response

Point 1: The authors claimed the method can deal with unpaired settings, but the experiments are based on paired ones. The authors can construct unpaired datasets based on existing ones. For example, paired (A1, B1), (A2, B2) can be reconstructed as (A1, B2), (A2, B1).

Response 1: Thank you for your proposal. The method in this paper is trained on paired data sets, but through the two GANs and the loss function combining adversarial loss and perceptual loss, the model trained in this article has the ability to deal with unambiguous data sets. And in the experimental results, this paper not only tested on the paired datasets, but also tested on the self-built unpaired datasets. The experimental results show that our method can handle unpaired datasets.

Point 2: To my knowledge, training GAN is unstable. How to tackle this in this paper? Using cycle-consistent GAN might be a solution.

Response 2: The model in this paper is based on CycleGAN, and combines the loss function of Wasserstein distance and the combination of adversarial loss and perceived loss to achieve a stable effect.

Point 3: The ablations studies are missing on each component of the method.

Response 3: In 346. Ablations studies have been performed for each component of the method.

Point 4: Some closely related works are missing, such as "Text image deblurring via two-tone prior", "Towards more efficient and flexible face image deblurring using robust salient face landmark detection" on image deblurring.

Response 4: line 116. The above literature has been cited in this paper.

Point 5: Careful proofreading is required.

Response 5: The paper has been carefully proofread.

Round 2

Reviewer 2 Report

The authors have addressed my previous concerns. I do not have other concerns and thus recommend acceptance.